# Advanced Glycation End Products of Bovine Serum Albumin Suppressed Th1/Th2 Cytokine but Enhanced Monocyte IL-6 Gene Expression via MAPK-ERK and MyD88 Transduced NF-κB p50 Signaling Pathways

**DOI:** 10.3390/molecules24132461

**Published:** 2019-07-04

**Authors:** Chieh-Yu Shen, Cheng-Han Wu, Cheng-Hsun Lu, Yu-Min Kuo, Ko-Jen Li, Song-Chou Hsieh, Chia-Li Yu

**Affiliations:** 1Division of Rheumatology, Immunology and Allergy, National Taiwan University Hospital, National Taiwan University College of Medicine, Taipei 10002, Taiwan; 2Institute of Clinical Medicine, National Taiwan University College of Medicine, Taipei 10002, Taiwan; 3Institute of Molecule Medicine, National Taiwan University College of Medicine, Taipei 10002, Taiwan

**Keywords:** advanced glycation end products, N^ε^-(carboxymethyl)-lysine, Th1/Th2 cytokines, IL-6, MAPK-ERK1/2, MyD88, NF-κB p50, inflamm-aging

## Abstract

Advanced glycation end products (AGE), the most known aging biomarker, may cause “inflamm-aging” (i.e., chronic low-grade inflammation that develops with aging) in both aged and diabetes groups. However, the molecular bases of inflamm-aging remain obscure. We prepared AGE by incubating BSA (0.0746 mmol/L) + glucose (0.5 mol/L) at 37 °C in 5% CO_2_–95% air for 1–180 days. The lysine glycation in BSA–AGE reached 77% on day 30 and 100% after day 130, whereas the glycation of arginine and cysteine was minimal. The N^ε^-(carboxymethyl)-lysine content in BSA–AGE was also increased with increasing number of incubation days. The lectin-binding assay revealed that the glycation of BSA not only altered the conformational structure, but lost binding capacity with various lectins. An immunological functional assay showed that BSA–AGE > 8 μg/mL significantly suppressed normal human Th1 (IL-2 and IFN-γ) and Th2 (IL-10) mRNA expression, whereas AGE > 0.5 μg/mL enhanced monocyte IL-6 production irrelevant to cell apoptosis. The AGE-enhanced monocyte IL-6 production was via MAPK–ERK and MyD88-transduced NF-κBp50 signaling pathways. To elucidate the structure–function relationship of BSA–AGE-enhanced IL-6 production, we pre-preincubated BSA–AGE with different carbohydrate-degrading, protein-degrading, and glycoprotein-degrading enzymes. We found that trypsin and carboxypeptidase Y suppressed whereas β-galactosidase enhanced monocyte IL-6 production. In conclusion, BSA–AGE exerted both immunosuppressive and pro-inflammatory effects that are the molecular basis of inflamm-aging in aged and diabetes groups.

## 1. Introduction

Advanced glycation end products (AGE), a deleterious post-translated non-enzymatic modification of macromolecules (proteins, lipids, and nucleic acids) by sugars is increased in the plasma obtained from the aged [1,2,3,4,5,6] and in those with related neurodegenerative diseases [7,8,9], diabetes and related complications [10,11,12,13,14,15,16,17,18,19,20], and autoimmune/inflammatory diseases such as systemic lupus erythematosus (SLE), rheumatoid arthritis (RA), progressive systemic sclerosis (PSS), adult-onset Still’s disease, and chronic inflammatory psoriasis [21,22,23,24,25,26,27,28,29]. The glycation of macromolecules is induced by the Maillard reaction, which includes the slow formation of a Schiff base, early reversible AGE by Amadori rearrangement, and finally late irreversible AGE as shown in Figure 1A [6,30]. Alternatively, AGE can be formed more rapidly via the intermediate formation of reactive carbonyl compounds such as methylglyoxal or glyoxal in circumstances of oxidative and carbonyl stresses [31]. Certain pathological conditions such as hyperglycemia [15,16], increased oxidative stress [6,9,32,33], decreased deglycant capacity [1], and increased dietary AGE intake can initiate AGE formation [34]. Increased plasma levels of AGE can elicit vascular damage via increased leukocyte adhesion, vascular permeability, contracting basement membrane thickening, endothelial cell apoptosis, matrix accumulation, abnormal angiogenesis, and inflammation [10,12]. AGE induce vascular damage in patients with aging and diabetes mellitus by binding with their receptor RAGE, that may enhance intracellular oxidative stress [10,18] and activating RAS–MAPK (p38 and ERK1/2)–NF-κB and AP-1 pathways [12,13,17,35,36,37,38].

Aging is a complex multifactorial process characterized by the accumulation of deleterious changes in different cells and tissues that lead to the progressive deterioration of structural integrity and physiological functions in multiple organ systems [1,2,3,4]. It is conceivable that macromolecular damage and biochemical changes may occur in physiological ageing and age-related disorders including diabetes, cataract, Alzheimer’s disease, amyloidosis, atherosclerosis, and Parkinson’s disease [1]. Accordingly, ageing and age-related disorders can be attributed to the process of macromolecular glycation as a common event. Semba et al. [2] concluded that the accumulation of AGE-modified macromolecules either ingested in foods or upregulated by inflammation may accelerate the multisystem function decline in ageing and therefore contribute to the ageing phenotypes. Simm [37] further demonstrated that glycation can modify the structure and function of intracellular and extracellular matrix expressions to cause tissue stiffness and long-lasting inflammatory response. The increased formation and accumulation of AGE in aged and diabetes groups can cause cardiovascular and neurodegenerative complications [1,2,3,4,36,37,38]. Interestingly, the combination of inflammation and immunosenescence as “inflamm-aging” [39,40] is a common feature in the two groups [39,40,41,42,43]. Nevertheless, the real mechanisms for inflamm-aging remain undetermined. Changes in the innate and adaptive immune responses, chronic antigen stimulation, and the appearance of endogenous macromolecule and senescent cells in the blood are potential contributing factors for inflamm-aging [39,40]. In the present study, we sought to investigate the effects of bovine serum albumin (BSA)–AGE on immune responses and inflammation, and attempted to elucidate the molecular basis of inflamm-aging.

## 2. Results and Discussion

### 2.1. Formation, Changes of Color and Molecular Weight, and Amino Acid Residue Glycation in BSA–AGE

During incubation of BSA (0.0746 mmol/L) + glucose (0.5 mol/L) mixture at 37 °C for 1–180 days, we noted that the color of the mixture gradually changed from transparent to a light yellow at day 45, orange at day 130, and finally became brown at day 180 with slightly viscose, as shown in Figure 1B. The molecular weight of these BSA–AGEs was analyzed by 10% SDS-PAGE. We found that the relative molecular weight of BSA–AGEs gradually increased upon the progression of glycation (Figure 1C). For further confirmation of the formation of BSA–AGEs, we measured the N^ε^−(carboxymethyl)-lysine (CML) contents, a representative product of AGE formation, in different BSA–AGEs by ELISA. As shown in Table 1, the amount of CML in BSA–AGE increased in parallel to incubation days. In addition, the lysine glycation of BSA–AGEs reached 77% on day 30 and was maximal after day 130 as detected by LC/MS (Table 1). Nevertheless, arginine glycation was modest and there was no glycation in the cysteine residues of BSA–AGE. The glycation sites in lysine and arginine of three BSA–AGEs are shown in Appendix A. The detailed procedures of sample preparations, LC/MS proteomics measurement, and data interpretation are presented in Appendix A as a file. Based on these results, we used the latest stable day 180 BSA–AGE in the forward experiments. It has been demonstrated that the amino acid residues most susceptible to glycation in albumin are lysine, arginine, and cysteine [44]. Lysine-525 has been reported as a dominant site for glycosylation in human albumin [45,46]. However, we found the three lysine residues—K-431, K-439, and K-533—in BSA that could not be glycated (Appendix A), which is different from human serum albumin. The glycation of amino acids induces several structural modifications, including to molecular weight, and alterations in the secondary and tertiary chemical structures [47,48,49]. In addition, the glycation-induced structural modifications have determinant impacts on albumin functions. The anti-oxidant effects of albumin are diminished [50,51], whereas the binding affinity of glycated albumin with other molecules remains unchanged [52]. Furthermore, the glycation of albumin can alter glucose metabolism in skeletal muscle and adipose tissue [53,54]. The increased expressions of CML in people of different ages and in whose skin has been subjected to different levels of sun exposure indicates that the presence of protein glycation may become a biomarker of the aging process in humans [55].

### 2.2. The Lectin-Binding Capacity of Different BSA–AGEs

The specific binding of lectins to particular sugar moieties can be applied to identify the terminal carbohydrate structures in the molecules. We detected the lectin-binding capacity of BSA–AGEs from day 1 to day 180 using the digoxigenin (DIG)-labeled glycan differentiation kit. MAA (*Maackia amurensis* agglutinin) identifies sialic acid terminally linked [2-3] to galactoses. SNA *(Sambucus nigra* agglutinin) identifies sialic acids linked [2-6] to galactose and sialic acid in O-glycan structures. This lectin is also suitable for complex sialylated N-glycan chains in combination with MAA. DSA (*Datura stramonium* agglutinin) identifies Gal-[1-4]GlcNAc in complex and hybrid N-glycans. PNA (peanut agglutinin) identifies the core galactose [1-3]N-acetylgalactosamine and is thus suitable for identifying O-glycosidically linked carbohydrate chains. As demonstrated in Figure 2, BSA per se can conjugate with MAA (Figure 2A), SNA (Figure 2B), and DSA (Figure 2C), but not PNA, (Figure 2D). However, the binding capacity with the former three lectins gradually diminished in parallel to the number of lysine glycations in day 130 and day 180 BSA–AGEs. These results may indicate that BSA molecule per se contains a complex carbohydrate structure capable of binding with different lectins. The glycation sites of the BSA molecule change the conformational structures and consequently diminish the binding capacity with different lectins.

### 2.3. Dose-Dependent Effects of BSA–AGE on Th1 (IL-2 and IFN-γ) and Th2 (IL-10) Cytokine mRNA Expression, and Monocyte IL-6 Production by Anti-CD3 + Anti-CD28 Activated Normal Human Mononuclear Cells

AGE have been reported to be capable of activating immune and inflammatory responses [56,57,58] and endothelial/epithelial cells [59,60,61] via binding to their receptors (RAGE—receptor of advanced glycation end products) [56,57,60,62,63]. Many authors have demonstrated that RAGE is a multi-ligand receptor for AGE, calgranulin, s100 protein, and high-mobility group box1 (HMGB1) [23,56,64,65]. We detected the effects of BSA–AGE (1–16 μg/mL) on Th1 (represented by IL-2 and IFN-γ) and Th2 (represented by IL-10) mRNA expressions and monocyte IL-6 production by activated mononuclear cells (MNCs). We are the first group to demonstrate that BSA–AGE dose-dependently suppressed Th1 and Th2 cytokine mRNA expression (Figure 3A). The immunosuppressive effects of BSA–AGE had not been reported in the literature. In contrast, the IL-6 production was conversely enhanced to a maximum of 6 μg/mL, which is compatible with the previous reports [56,58,61] (Figure 3B). We then measured the percent cell apoptosis of MNCs after incubation with different amounts of BSA–AGE. As demonstrated in Figure 3C, it was found that cell apoptosis was increased, but not to statistical significance, by high concentrations (4–16 μg/mL) of AGE. For further confirmation that BSA–AGE actually enhances monocyte IL-6 production, we compared the same concentration of BSA–AGE and BSA in terms of IL-6 production. As shown in Figure 4A, BSA–AGE tremendously enhanced IL-6 production by activated MNC. These results indicate that AGE exerts both immunosuppressive and proinflammatory effects on the immune system that are the characteristics of inflamm-aging in the aged and diabetic populations. Although a few authors reported that IL-6 production by activated MNCs in the aging group did not differ from that in the young group [66,67], many reports have demonstrated that IL-6, TNF-α, and IL-1β are biomarkers for aging [68,69,70,71]. Interestingly, IL-6 production is also increased in diabetes patients with cardiovascular and renal complications [72,73,74,75]. These results suggest that AGE exert both immunosenescent and pro-inflammatory effects on immune and vascular systems that elicit inflamm-aging effects in both aged and diabetic groups. It is conceivable that interleukin-6 is a growth and differentiation factor for B cells, Th17 cells, and cytotoxic T cell response to infections [76]. IL-6 works synergistically with IL-3 in hematopoiesis and becomes a major inducer for acute-phase reaction [77]. IL-6 also plays an important role in skeletal homeostasis and contributes to osteoporosis [78]. A number of studies have investigated the response of IL-6 after AGE stimulation and showed that elevated IL-6 production induced active inflammation [79,80]. As a brief conclusion, the dissociation of lectin-binding alterations (Figure 2) and immunological effects (Figure 3) of BSA–AGE were found. We speculate that the glycation of amino acid residues (i.e., lysine, arginine, and/or cysteine) in BSA alters the physical confirmation and glycan chemistry that are crucial for lectin-binding but not for receptors (RAGEs) binding capacity.

### 2.4. MAPK-ERK and MyD88 Transduced NF-κB Signaling Pathways are Involved in BSA–AGE-Enhanced IL-6 Production by Activated Normal MNCs

To identify the signaling pathway(s) involved in AGE-enhanced IL-6 production, anti-CD3+Anti-CD28 activated MNCs were pre-incubated with different protein inhibitors. We found that PD98059 (50 μmol/L, a non-competitive ERK inhibitor) and NBP2-29327 (100 μmol/L, a MyD88 inhibitor) significantly suppressed AGE (0.5 μg/mL)-enhanced monocyte IL-6 production. However, SB203580 (1 μmol/L, a p38 MAPK inhibitor) or wortmannin (0.1 μmol/L, a specific PI3K inhibitor) had no effect on BSA–AGE-enhanced IL-6 production (Figure 4B). For further confirmation that the MAPK-ERK signaling actually mediated AGE-enhanced IL-6 production, we pre-treated MNCs with PD98059 and then reacted with different concentrations of AGE (0.5–12 μg/mL). As shown in Figure 4C, the ERK inhibitor significantly suppressed AGE-enhanced IL-6 production. Since both ERK and MyD88 are the upstream signaling molecules of NF-κB nuclear transcription, we then measured the amount of NF-κB components p50 and p65 in activated MNCs in the presence of BSA–AGE (1–16 μg/mL). We noted that only NF-κBp50, but not p65, was increased in the presence of 1 and 4 μg/mL BSA–AGE (Figure 4D). Our results seem to be compatible with the reports that AGE-LDL-enhanced IL-6 production by epithelial cells occurred via binding to the TLR2/4-MyD88-NF-κB signaling pathway [81,82]. In brief conclusion, we found that both MAPK-ERK and MyD88 transduced NF-κB p50 signaling pathways were involved in BSA–AGE-enhanced IL-6 production by activated MNCs. However, our results are not consistent with other authors, who found that MAPK-p38 and PI3K were involved in the AGE-induced inflammatory mechanism [12,13,17,31,34,35]. This may be due to use different kinds of cells in the experiments.

### 2.5. Protein- and Glycoprotein-Degrading Enzymes Decrease Whereas a Particular Glycome-Modifying Enzyme Enhances IL-6 Production by Activated MNCs

To explore the intra-molecular domain structure in BSA–AGE responsible for monocyte IL-6 enhancing activity, we treated AGE with carbohydrate-degrading enzymes including α-glucosidase, β-galactosidase, and neuraminidase; protein-degrading enzymes including trypsin and proteinase K; and the glycoprotein-degrading enzyme carboxypeptidase Y. We found that BSA–AGE-enhanced IL-6 production was remarkably diminished by trypsin and carboxypeptidase Y (Figure 5). The degree of suppression by trypsin digestion was greater than that by carboxypeptidase Y digestion. This may be because the size of peptide–glucose conjugates after trypsin digestion was much smaller than those after carboxypeptidase Y digestion. In addition, Pan et al. [83] demonstrated that trypsin cleaves the C-terminus of the protein in lysine and arginine residues, which are the important sites for glycation in BSA molecules. On the other hand, AGE digested by β-glucosidase or neuraminidase did not change their IL-6 enhancement (data not shown).

Unexpectedly, β-galactosidase-treated BSA–AGE conversely enhanced IL-6 production (Figure 5). In light of the DSA-binding capacity of BSA (Figure 2A), this may suggest that BSA per se contains galactose in the structure of Gal-[1-4]GlcNAc that forms the complex and hybrid N-glycan. These intra-molecular galactoses can also link to sialic acids and then bind to MAA (Figure 2A). We speculate that the galactose-moiety-related O-glycan structure in BSA per se can impair IL-6 production. After removing the inhibitory O-glycan structure by galactosidase, increased IL-6 production by the modified BSA–AGE appeared. The new emerging glycome constitutions and properties of BSA–AGE after β-galactosidase treatment are now under investigation. These results may suggest that a basal size of AGE–glucose conjugates and particular glycome constitutions in the BSA–AGE molecule are crucial for its IL-6-enhancing activity.

Figure 6 summarizes the effects of BSA–AGE on inflamm-aging and their potential signaling pathways. Figure 6A shows the effects of BSA–AGEs on immunosuppression, inflammation, and vasculopathy. Figure 6B presents the possible signaling pathways related to BSA–AGE-enhanced monocytes’ IL-6 production. In Figure 6C, a possible signaling pathway involving AGE-mediated endothelial cell damage is shown. The molecular basis of BSA–AGE-induced immunosenescence will be investigated in the near future.

## 3. Materials and Methods

### 3.1. Preparation of BSA–AGE

Bovine serum albumin (BSA) at a concentration of 0.0746 mmol/L was mixed with 0.5 mol/L of *D*-glucose in PBS (pH 7.4) in a sterile condition. The mixture was incubated in 5% CO_2_–95% air at 37 °C for 1–180 days. BSA alone was also incubated in the same condition (non-glycated BSA) as a negative control.

### 3.2. Identification of AGE

#### 3.2.1. Observation of Color Changes

The color of the mixture changed from being transparent (before day 30), to light red (between day 45 and day 130), and finally brown (day 180) with a slight viscosity. The non-glycated BSA remained clear and colorless.

#### 3.2.2. Changes of Relative Molecular Weight of BSA–AGE Detected by 10% SDS-PAGE

BSA–AGEs at different incubation times from day 1 to day 180 were electrophoresed in 10% SDS-PAGE. The dispersed molecules were stained by Coomassie blue to identify their relative molecular weight.

#### 3.2.3. Measurement of N^ε^-(Carboxymethyl)-lysine (CML) Content in BSA–AGE Molecule

We used the commercially available CircuLex CML ELISA Kit (CycLex Co., Ltd. Nagano, Japan) and followed the manufacturer’s instructions to quantify the amounts of CML adducts in different BSA–AGEs. Briefly, biochemically modified CML–BSA was pre-coated in microwells. Standards or samples and anti-CML adduct monoclonal antibody MK-5A10 was added for competition. After incubation, the amount of CML–BSA was calculated by the standard curve. The detection range of CML by this ELISA kit is 0.109–7.0 μg/mL.

#### 3.2.4. Detection of the Number of Glycation Sites in Lysine, Arginine, and Cysteine in Different BSA–AGEs by LC/MS Proteomics Analysis

We sent the samples at day 30, day 130, and day 180 to Mithra Biotechnology Inc. (New Taipei City, Taiwan) for LC/MS identification of amino acid glycation sites and glycation number in different BSA–AGEs. The procedures for data interpretation are provided as Appendix A.

Ammonium bicarbonate, dithiothreitol (DTT), iodoacetamide (IAM), and formic acid (FA) were purchased from Sigma-Aldrich (St. Louis, MO, USA). Acetonitrile (ACN) was purchased from Spectrum (New Brunswick, NJ, USA). Trypsin and chymotrypsin were purchased from Promega (Madison, WI. USA). RapiGest SF was purchased from Waters (Milford, MA, USA). Trifluoroacetic Acid (TFA) was purchased from Alfa Aesar (Ward Hill, MA, USA).

The three protein samples were buffer exchanged with 50 mM ammonium bicarbonate using a 10 kDa molecular weight cutoff filter. Then, 1% RapiGest SF surfactant was added to a final concentration of 0.1%, reduced with 5 mM DTT at 60 °C for 30 min, and then alkylated with 15 mM IAM in the dark at room temperature for 30 min. The resulting protein was digested with two enzymes with the following digestion conditions: (1) trypsin digestion at 37 °C overnight (protein:enzyme = 50:1), (2) chymotrypsin digestion at room temperature overnight (protein:enzyme = 50:1). After digestion, each sample was acidified with 0.5% (*v*/*v*) TFA as the final concentration, which was performed at 37 °C for 30 min and then centrifuged at 14,000 rpm for 30 min to precipitate the hydrolytic RapiGest SF by-product.

The digested samples were analyzed with a Q-Exactive mass spectrometer (Thermo Scientific, Waltham, MA, USA) coupled with an Ultimate 3000 RSLC system (Dionex, Sunnyvale, CA, USA). The LC separation was performed using a C18 column (Acclaim PepMap RSLC, 75 μm × 150 mm, 2 μm, 100 Å). Mobile phase A was 0.1% FA in H_2_O, and mobile phase B was 0.1% FA in 95% ACN. The flow rate was 0.25 μL/min. The gradient setting was 0 min-1%, 5.5 min-1%, 45 min-25%, 48 min-60%, 50 min-80%, 65 min-1%, 70 min-1%.

Full MS scan was performed with the *m*/*z* range of 150–2000, and the ten most intense ions from the MS scan were subjected to fragmentation for MS/MS spectra. Raw data were processed into peak lists by Proteome Discoverer 1.4 for Mascot database search.

Database search was performed with Mascot 2.4.1. with the instrument type as ESI-TRAP. We used the BSA Database: BSA. The fixed modification was Carbamidomethyl (C) and the variable modifications were Glycation (K), Glycation (A), and Glycation (C). The peptide mass tolerance was 10 ppm and the fragment mass tolerance was 0.05 Da. The ion cut-off score was 13.

#### 3.2.5. Detection of the Lectin-Binding Capacity of Different BSA–AGE Molecules

We used a DIG Glycan Differentiation Kit (Roche Diagnostics GmbH Roche Applied Science, Mannheim, Germany). Lectins including MAA, SNA, DSA, and PNA were used following the procedures suggested by the manufacturer. The positive control molecules were fetuin for MAA, SNA, and DSA, and asialofetuin for PNA. Briefly, various BSA–AGEs were loaded in 10% SDS-PAGE for electrophoretic analysis. After being electrotransferred to nitrocellulose paper, the dispersed molecules were reacted with digoxigenin-labeled lectin and then alkaline-phosphatase-conjugated anti-digoxigenin antibody. The complexes were finally color developed by a staining solution containing NBT/BCIP solution.

### 3.3. Isolation of Mononuclear Cells from Peripheral Blood of Normal Human Volunteers

This study was approved by the IRB and Ethical Committee, National Taiwan University Hospital, Taipei, Taiwan (201401102RIN). Informed consent was obtained from each participant. Heparinized venous blood was mixed with one-fourth volume of 2% dextran solution (mol. wt. 464,000 daltons) (Sigma-Aldrich Chemical Company, St. Louis, MO, USA) and incubated at room temperature for 30 min. Leukocyte-enriched supernatant was collected and layered on a Ficoll-Hypaque density gradient solution (specific gravity 1.077) (Pharmacia Biotech, Uppsala, Sweden) followed by centrifugation at 250× *g* for 25 min. The MNCs were aspirated from the interface. The cell concentration of MNCs was adjusted to 2 × 10^6^/mL in 10% fetal bovine serum in RPMI-1640. The MNC suspension contained 90% lymphocytes, 5%–8% monocytes, and 2%–5% other cell populations confirmed by Giemsa stain with a viability greater than 95%, as confirmed by trypan blue dye exclusion.

### 3.4. Incubation of MNCs with Different Concentrations of Day-180 AGE

Anti-CD3 and antiCD28 pre-activated MNCs (2 × 10^6^/mL) were incubated at 37 °C with 1, 4, 8, and 16 μg/mL of AGE for 2 h. The total mRNA was extracted for detection of the Th1 (denoted by IL-2 and IFN-γ) and Th2 (denoted by IL-10) cytokine mRNA expression. In addition, the pre-activated MNCs were incubated with different amounts of BSA–AGE for 24 h. The culture supernatants were collected for measuring the IL-6 production by ELISA.

### 3.5. Detection of Th1/Th2 Cytokine mRNA Expression

The cells were harvested after 2 h incubation of MNCs with BSA–AGE. The total cellular RNAs were extracted using the Ultraspec RNA isolation system (BIOTEX Lab., Inc. Houston, TX, USA). One microgram of RNA was reversely transcribed into cDNA in 30 µL of reverse-transcription buffer containing 50 mmol/L Tris–HCl, 75 mmol/L KCl, 3 mmol/L MgCl_2_, 0.5 µg oligo-dT primer, 0.5 mmol/L deoxynucleotide triphosphate (dNTP), 30 U RNasin, 10 mmol/L dithiothreitol (DTT), and 400 U murine Moloney leukemia virus (M-MLV) reverse transcriptase (Promega Corp., Madison, WI) in pH 8.3 at 42 °C for 1 h. After reverse transcription, 5 µL of product was added to PCR buffer containing 10 mmol/L Tris–HCl, 1.5 mmol/L MgCl_2_, 50 mM KCl, 0.1% Triton-X-100, 100 ng forward primer, 100 ng reverse primer, 0.2 mM dNTP, 2U DNA polymerase (Finnzymes Oy, Riihitontuntie, Finland), and 5% DMSO. The PCR was performed in a HYBAID OminGene DNA Thermal Cycler (Teddington, UK) with a program of denaturing at 95 °C for 45 s, annealing at 60 °C for 45 s, and primer extension at 72 °C for 2 min. The amplification was carried out for 30 cycles. The reaction was stopped in a final extension at 72 °C for 10 min, followed by incubation at 25 °C. The forward and reverse primers for human IL-2, IL-10, and IFN-γ were purchased from Clontech Laboratories (Palo Alto, CA, USA).

Oligonucleotide pair primers for different cytokine mRNA detections are shown in Appendix A.

The cDNA fragments amplified by these sets of primers were 123 bp for IL-2, 104 bp for IFN-γ, and 141 bp for IL-10. The PCR products were electrophoresed in 1.8% agarose gel with φx174 digested by HaeIII enzyme as a DNA calibration marker.

### 3.6. Detection of IL-6 Concentration in the Culture Supernatants by ELISA

The 24-h culture supernatants of MNCs incubated with different concentrations of BSA–AGE were collected for IL-6 determination by commercially available ELISA kits obtained from eBioscience (San Diego, CA, USA). The minimum detection concentration of IL-6 is 2 pg/mL.

### 3.7. Detection of Cell Apoptosis by ELISA

We used the annexin V-FITC apoptosis kit (BD Pharmagen, La Jolla, USA) to detect cell apoptosis. Briefly, MNCs (10^6^ cells/mL) were incubated with different concentrations of BSA–AGE for 24 h. After three washes, the cells were stained in the dark with annexin V and propidium iodide for 15 min. Then, the cell suspension was detected by FACsort flow cytometer (Becton-Dickinson, Mountain View, CA, USA). Annexin V-FITC and propidium iodide were recorded by FL1-H (525 nm) and FL2-H (575 nm) filters respectively.

### 3.8. Identification of Signaling Pathway(s) of BSA–AGE-Enhanced Monocyte IL-6 Production by Anti-CD3 + Anti-CD28 Antibodies-Activated Normal MNCs Using Different Protein Inhibitors

Human mononuclear cells (2 × 10^6^/mL) were pre-incubated with different signaling molecule inhibitors including PD98059 (50 μmol/L, a non-competitive ERK inhibitor), SB203580 (1 μmol/L, a MAPK-p38 inhibitor), or wortmannin (0.1 μmol/L, a specific inhibitor of PI3K) for 1 h, or with NBP2-29327 (100 μmol/L, a MyD88 inhibitor) for 24 h. After several washes, the MNCs were then activated with anti-CD3 + anti-CD28 antibodies in the presence of different concentrations of BSA–AGE for 24 h. The IL-6 concentrations in the culture supernatants were measure by a commercially available ELISA kit (BD Bioscience).

### 3.9. Detection of Intranuclear NF-κB Components, p65 and p50, by ELISA

Activated MNCs were cultured with different concentrations of AGE for 2 h. The nuclear extract was obtained by NE-PER nuclear and cytoplasmic extraction kits (Pierce Biotechnology, Rockford, IL, USA). Briefly, cells were washed twice in chilled PBS and then suspended in 500 μL of lysis buffer containing 10 mmol/L HEPES (pH 7.9), 50 mmol/L NaCl, 1 mM EDTA, 5 mmol/L MgCl, 10 mmol/L sodium orthovanadate, 10 mmol/L sodium molybdate, 0.2 mmol/L phenylmethylsulfonyl fluoride, 10 μg/mL protease inhibitors (pepstatin, leupeptin, aprotinin), and 0.05% Nonidet P-40. After incubation for 20 min on ice, glycerol was added (final concentration 5%, *v*/*v*) and nuclei were pelleted by centrifugation at 600 rpm for 10 min at 2 °C. After rapid washing with the same buffer (containing 140 mmol/L NaCl instead of 50 mmol/L), the nuclei were gently resuspended in 100 μL of storage buffer containing 10 mM HEPES (pH 7.9), 400 mmol/L NaCl, 0.1 mmol/L EDTA, 10 mmol/L dithiothreitol, 5% glycerol, 0.5 mmol/L phenylmethyl-sulfonyl fluoride, 10 mmol/L sodium vanadate, 10 mmol/L sodium molybdate, and 10 μg/mL protease inhibitors. After 60 min on ice with gentle mixing, particulate matter was eliminated by centrifugation at 16,000× *g* for 10 min at 4 °C. Protein content in the supernatant (the extracted nuclear proteins) was determined using Bradford’s method. The amount of NF-κB p50 and p65 protein was determined by commercially available ELISA kits obtained from Cayman Chemical Co. (Ann Arbor, MI, USA).

### 3.10. Detection of the Effects of Carbohydrate-, Protein-, and Carbohydrate-and-Protein-Degrading-Enzyme-Digested BSA–AGE on IL-6 Production by Anti-CD3 + Anti-CD28 Activated Normal MNCs

We treated AGE with 1 μg/mL of different carbohydrate-degrading enzymes including α-glucosidase, β-galactosidase, and neuraminidase; protein-degrading enzymes including trypsin and proteinase K; or glycoprotein-degrading enzyme carboxypeptidase Y for 1 h. The digested BSA–AGE was then added to an activated-MNC suspension (2 × 10^6^/mL) for 24 h. We also co-cultured resting MNCs with the same concentration of different enzymes for 24 h as a negative control. The collected supernatants were detected for IL-6 concentration.

### 3.11. Statistical Analysis

All of the results were represented as mean ± s.d. Continuous variables were assessed by non-parametric Wilcoxon rank-sum test using a commercially available software package (SPSS for Windows, Version 16.0, SPSS Inc. Chicago, IL, USA). A *p*-value < 0.05 was considered statistically significant.

## 4. Conclusions

BSA–AGE exerted both immunosuppressive and inflammatory effects on the human immune system via MAPK-ERK and MyD88 transduced NF-κBp50 signaling pathways. The IL-6-enhancing activity of BSA–AGE depended on the basal size of peptide–glucose conjugates and the particular glycome constitutions of the molecules. These findings can explain the inflamm-aging phenomenon commonly found in aged and diabetic groups. Although we used BSA–AGE in the present study, human serum albumin (HSA)–AGE would be more logical and preferable in clinical studies. We are now preparing HSA–AGE for human immunosenescence studies.

## Figures and Tables

**Figure 1 molecules-24-02461-f001:**
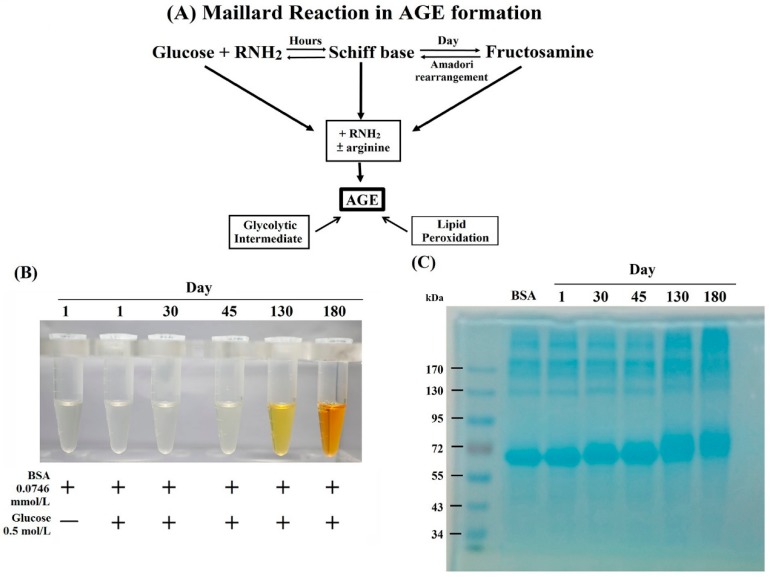
(**A**) Maillard reaction in advanced glycation end products (AGE) formation. (**B**) Change of color in bovine serum albumin (BSA)–AGE was observed from transparent to light yellow at day 45, orange color at day 130, and finally brown at day 180 with slightly viscose. (**C**) Estimation of AGE molecular weight by 10% SDS-PAGE.

**Figure 2 molecules-24-02461-f002:**
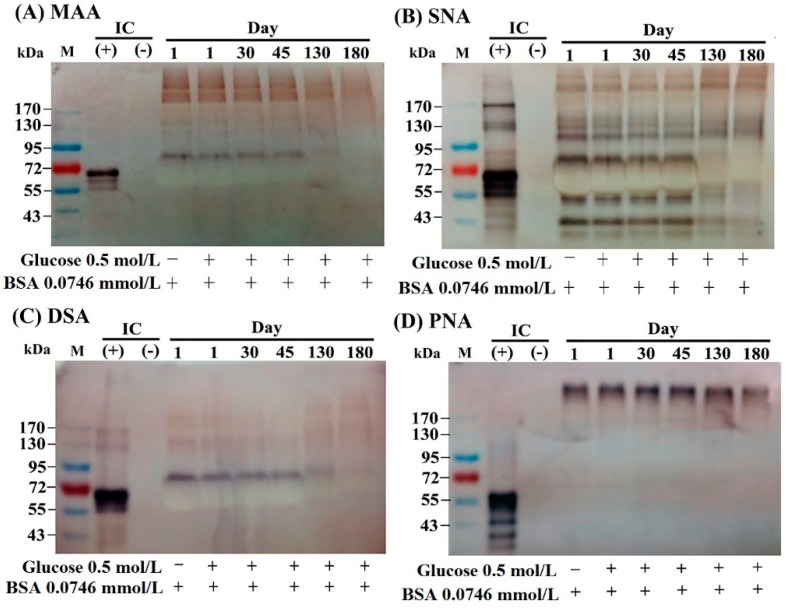
Changes in the lectin-binding capacity of different AGE as detected by a DIG glycan differentiation kit. (**A**) Binding capacity with MAA (*Maackia amurensis* agglutinin) that recognizes sialic acid linked [2-3] to galactose and also identifies -linked sialic acids in O-glycans. (**B**) Binding capacity with SNA (*Sambucus nigra* agglutinin) that recognizes sialic acid linked [2-6] to galactose in O-glycan structures. (**C**) Binding capacity with DSA (*Datura stramonium* agglutinin) that recognizes Gal-[1-4]GlcNAc in complex and hybrid N-glycan. (**D**) Binding capacity with PNA (peanut agglutinin) recognizes the core disaccharide galactose [1-3]N-acetylgalactosamine and is thus suitable for identifying O-galactosidically linked carbohydrate chains. GlcNAc: N-acetyl-glucosamine; IC: internal control.

**Figure 3 molecules-24-02461-f003:**
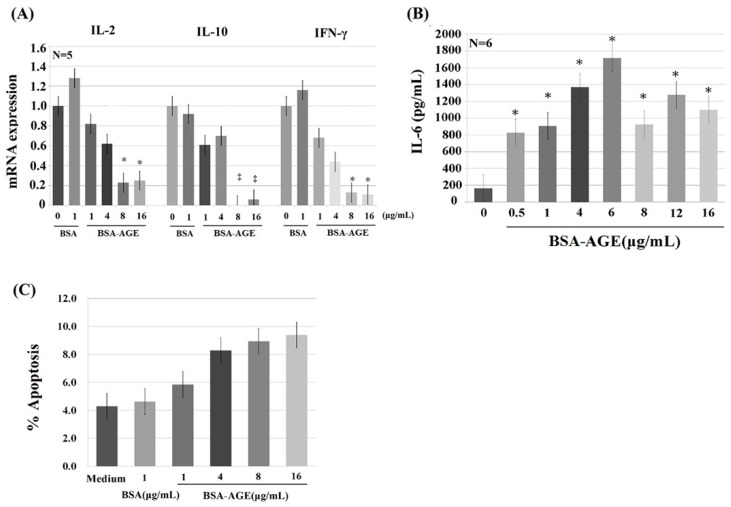
The dose-dependent effect of late AGE (day 180) on the gene expression of Th1 (represented by IL-2 and IFN-γ), Th2 (represented by IL-10), monocyte proinflammatory cytokine (represented by IL-6), and cell apoptosis. (**A**) Th1 and Th2 cytokine mRNA expression. (**B**) Monocyte IL-6 cytokine production. (**C**) Percent apoptosis of mononuclear cells. * *p* < 0.05 and ^‡^
*p* < 0.01 compared to BSA (1 μg/mL).

**Figure 4 molecules-24-02461-f004:**
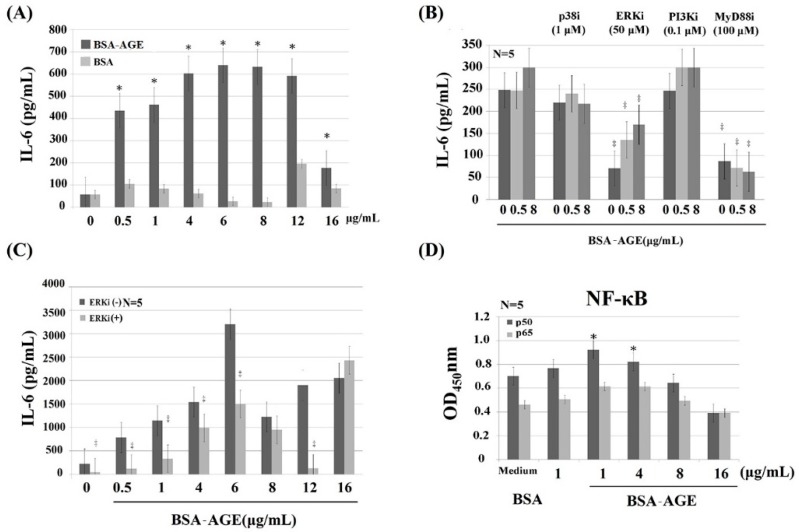
Dissection of signaling pathways of AGE-enhanced monocyte IL-6 production by using different signal molecule inhibitors. (**A**) Comparison of IL-6 production by different doses of AGE and BSA from 0.5–16 μg/mL. (**B**) Effects of different signaling molecule inhibitors on two doses (0.5 and 8 μg/mL) of AGE-enhanced IL-6 production. We used p38 inhibitor (SB203480, 1 μmol/L); ERK inhibitor (PD98059, 50 μmol/L); PI3K inhibitor (wortmannin, 0.1 μmol/L); and MyD88 inhibitor (NBP2-29327, 100 μmol/L). (**C**) The effect of ERK inhibitor (50 μmol/L) on different doses (0.5–16 μg/mL) of AGE-enhanced IL-6 production. (**D**) Effect of AGE (1–16 μg/mL) on the expression of NF-kB components p50 and p65. * *p* < 0.05 and ^‡^
*p* value < 0.01 compared to medium control.

**Figure 5 molecules-24-02461-f005:**
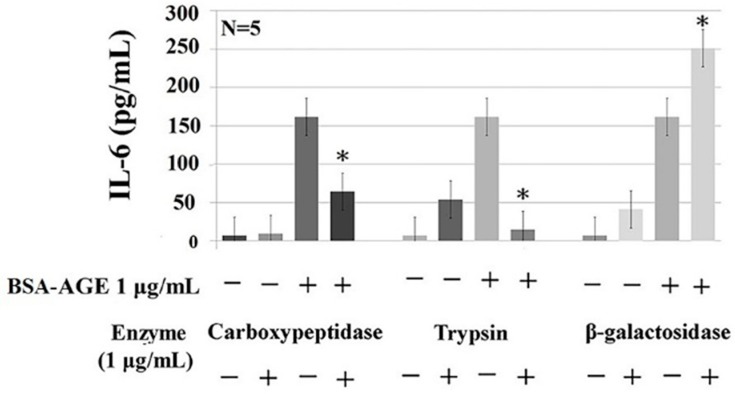
Effects of protein-degrading (trypsin, 1 μg/mL), carbohydrate-degrading (β-galactosidase, 1 μg/mL), and glycoprotein-degrading (carboxypeptidase, 1 μg/mL) enzymes on AGE-enhanced monocyte IL-6 production. * *p* value < 0.05 compared to AGE (1 μg/mL) only.

**Figure 6 molecules-24-02461-f006:**
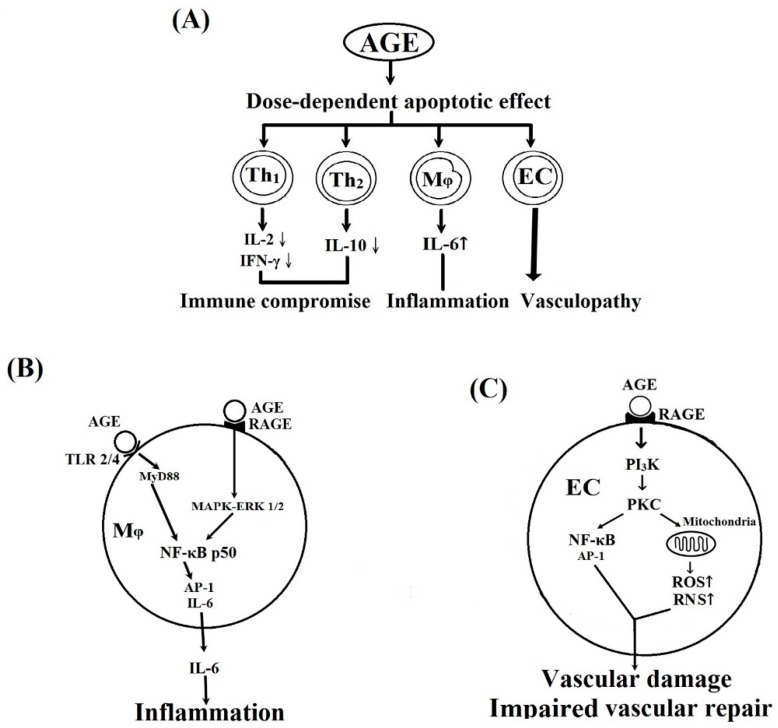
Schemes illustrate the effects of BSA–AGE on different cells and their possible signaling pathways. (**A**) The effects of AGE on Th1, Th2, monocyte/macrophages, and vascular endothelial cells (ECs). (**B**) The possible signaling pathways of AGE-enhanced IL-6 production by monocytes/macrophages and inflammation. (**C**) The possible signaling pathways of AGE on vascular endothelial cell and vascular damage. RAGE: receptor of AGE; RNS: reactive nitrogen species; ROS: reactive oxygen species.

**Table 1 molecules-24-02461-t001:** N^ε^(carboxymethyl)-lysine (CML) contents and % amino acid residue glycation in AGE-BSA molecules with different incubation time.

Incubation Time of AGE	N^ε^(carboxymethyl)-Lysine (μg/mL)			Number(%) of Glycation in AGE		
	Exp. 1	Exp. 2	Exp. 3	Lysine	Arginine	Cysteine
Day 1	0.056	0.049	—	—	—	—
Day 30	0.136	0.269	0.458	43/56(77%)	2/23(8%)	0/35(0%)
Day 45	1.126	0.171	0.186	—	—	—
Day 130	7.586	7.197	7.779	56/56(100%)	1/23(4%)	0/35(0%)
Day 180	10.916	10.664	9.509	56/56(100%)	4/23(16%)	0/35(0%)

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
