# Peer review of "Advanced Glycation End Products of Bovine Serum Albumin Suppressed Th1/Th2 Cytokine but Enhanced Monocyte IL-6 Gene Expression via MAPK-ERK and MyD88 Transduced NF-κB p50 Signaling Pathways"

_molecules, 2019, doi:10.3390/molecules24132461_

Round 1
Reviewer 1 Report
The submitted paper describes the preparation of AGE modified BSA (BSA-AGE) and the evaluation of the influence of BSA-AGE on the alterations of biochemical pathways. The submitted study is well organized and the contents of the study should provide useful information for readers. While, before the publication, several issues to be discussed are presented in the present manuscript. Please consider the revision of the manuscript according to the comments listed below.
(1) To evaluate the influence of the AGE modification of protein on human health, human serum albumin (HSA) should be more preferable than BSA. Please explain the reason why the authors selected BSA as a model protein, not HSA.
(2) The results of Figure 2 demonstrated that the AGE modification could diminish the binding capability of BSA to several lectins. However, the participation of the such diminished binding capacity of BSA-AGE to the alterations of biochemical pathways mentioned in Figure 3 and 4 was obscure. If possible, please explain the relationship between the diminishing of binding capacity of BSA-AGE and the alteration effect of BSA-AGE on mRNA expression and protein production.
(3) Although the caption of Figure 3-B was described as “IL-6 cytokine mRNA expression”, in actual, the content of Figure 3-B should be the effect on the IL-6 production (line 129 of page 5).
(4) It seemed that the protein production amount induced by BSA-AGE shown in Figure 3-B (800-1700 pg/mL) were significantly different with that shown in Figure 4-A (200-600 pg/mL) although the experimental design should be same.
(5) Figure 6-C illustrated the participation of ROS and NOS (maybe, RNS?) to the alterations of biochemical pathways induced by AGE modified protein. However, in the present manuscript, the participation of ROS was not mentioned at all.
Author Response
Answers to Reviewer [1]:
Question (1): To evaluate the influence of the AGE modification of protein on
human health, human serum albumin (HSA) should be more preferable
than BSA. Please explain the reason why authors selected BSA as a
model protein, not HSA.
Answer: As commented by the Reviewer, HSA-AGE would be more preferable than BSA-AGE in clinical sense. We are now gradually using HSA-AGE for further investigation. The reason why we used
BSA-AGE in the present investigation can be explained by the following 2 points:
(1) Chemical grade BSA was easily obtained rather than HSA around 4-5 years ago in Taiwan. We
decided to use BSA-AGE after literature review at that time by the fact that both BSA-AGE and HSA-AGE exerted equal effects on human PMN (Eur J Clin Invest 31: 1064-9, 2001)
and monocytes/macrophages (Cytokine 28: 35-47, 2004). For convenience, we followed the
methodology to prepare BSA-AGE for the present investigation.
(2) Recently published literatures also revealed that BSA-AGE exhibited biological effects on human cells and cell lines (J Biol Chem 290: 28189-99, 2015; Cell Physiol Biochem 43: 1571-87, 2017; J Priodontol Res 52: 268-76, 2017; Acta Diabetologica 55: 419-27, 2018).
Putting these results together, it may indicate that BSA-AGE can cross the interspecies barrier and can be applied in the studies of human and non-human cells. However, HSA-AGE would be more logical and suitable for human health/diseases as pointed out by the Reviewer. Accordingly, we have added a statement in “Conclusions” section (line 266) that;
“Although we used BSA-AGE in the present study, HSA-AGE would be logical and preferable in clinical studies. We will use HSA-AGE for human studies in future”.
Question (2): The results of Fig.2 demonstrated that AGE modification could diminish the binding capacity of BSA-AGE to several lectins. However, the participation of such diminished binding capacity of BSA-AGE to the alterations of biochemical pathway mentioned in Fig.3 and 4 was obscure. If possible, please explain the relationship between the diminishing of binding capacity of BSA-AGE and the alteration effect of BSA-AGE on Th1/Th2 mRNA expression and monocyte IL-6 protein production.
Answer: Thanks for this constructive suggestion. It is conceivable that lectins are proteins/glycoproteins capable of interacting with specific transformed glycan structures in the molecules. As we showed in Fig.2, BSA per se seems to contain the glycan structures that can bind with MAA, SNA and DSA, but not PNA. The glycation of amino acid residues in BSA not only alter the physical confirmation but the glycan chemistry. Therefore, the binding of BSA-AGE with different lectins diminished. However, the biological activities of BSA-AGE depend on the binding with their specific receptors, RAGE. On the other hand, RAGE can bind with HMGB-1, calgranulin, and S100 protein in addition to AGE. We speculate that glycation of lysine, arginine or cysteine residues do not change the crucial structures capable of binding with RAGE. Accordingly, the lectin-binding alterations shown in Fig.2 and the immunological effects shown in Fig.3 were dissociated. We have already added these statements in line 183 that;
“In short conclusion, the dissociation of lectin-binding alterations (Fig.2) and immunological effects (Fig.3) of BSA-AGE was found. We speculate that glycation of amino acid residues (lysine, arginine and/or cysteine) in BSA alters the physical confirmation and glycan chemistry that are crucial for lectin-binding but not for receptors, RAGEs, binding capacity.”
Question (3): Although the caption of Figure 3-B was described as “IL-6 cytokine mRNA expression”. In actural, the context of Figure 3-B should be the effect on IL-6 production (line 129 of page 5).
Answer: We appreciate Reviewer to point-out these errors. In the revised version, the ligand of the Fig. 3 has been changed to “gene expression” instead of “mRNA expression”. And (B) has been changed to “monocyte IL-6 cytokine production”.
Question (4): It seemed that the protein production amount induced by BSA-AGE shown in Figure 3-B (800-1700 pg/mL) were significantly different with that shown in Figure 4-A (200-600 pg/mL) although the experimental design should be same.
Answer: Thanks for pointing-out the significant inter-experimental difference in IL-6 production shown in Fig.3-B and Fig.4-A. It is our carelessness for not clear explanation in this “inter-experimental” difference. We used different blood mononuclear cells from different individuals in different times to assay the IL-6 production by using different lot number of IL-6 assay kit. Expectedly, a similar tendency in monocyte IL-6 production in Fig.3-(B) and Fig.4-(A) was derived.
Question (5): Fig.6-C illustrated the participation of ROS and NOS (maybe RNS?) to the alterations of biochemical pathways introduced by AGE modified protein. However, in the present manuscript, the participation of RNS was not mentioned at all.
Answer: We should apologize for the typing error in Fig.6-C. The “NOS” is absolutely a mistake of “RNS”. In fact, oxidative stress may also induce RNS production in addition to ROS. In fact, Fig.6-C was depicted according to the other authors’ results entirely for easy explanation of inflamm-aging mechanism in aged group.

Reviewer 2 Report
Dear Sir or Madam, the Manuscript „ Advanced Glycation Endproducts (AGEs) Suppress Th1/Th2 Cytokine but Enhance Monocyte IL-6 Gene Expression via MAPK- ERK and MyD88 transduced NF-κBp50 Signaling Pathways “ describes response of cultured human monocytes to a treatment with glycated BSA in vitro. The topic is very interesting; the data are rich and comprehensive. However, major revision is required, before this manuscript can be published.
Major remarks
1. English needs to be improved!
2. The largest problem of this work: why CML is taken for characterization of glycation levels? Why only CML? Why modifications at arginine residues are not considered? To some extent, proteomics studies could provide some information, but these data are not shown! This issue needs to be comprehensively addressed and improved. The authors also define the samples in a strange way.
3. Line 16 and everywhere as applicable: correct SI unit for concentration is mol/L, but not M. Please change everywhere appropriately. And think about spaces between values and units.
4. Line 25: what do you define as “AGE” here? AGE-modified protein? Needs to be precisely defined!
5. Introduction appears too short – it needs to be more detailed to more comprehensively disclose interrelations of AGEs and ageing.
6. I generally don’t understand, how ELISA was done here. Protocol and antibody description are necessary.
7. Section 4.1: with the progress of Maillard reaction, original protein solution typically becomes a bit viscous. That is why, an important question – did you shake incubations with HAS?
8. Section 4.2.4: the provided description of proteomics experiment is not sufficient. Please, provide exact and detailed procedure for sample preparation, measurements and data interpretation.
9. Section 4.2.5: the information, how lectin-binding test was done is missing
10. Section 4.5: too detailed with primers. Put primer sequences as a table in supplement
11. Please, make a list of abbreviations – it is really difficult without such a list in the case when results stand before methods.
12. Line 11: what do you mean as “different AGEs”? you quantify only CML – I would not extrapolate it to all AGEs.
Minor remarks
1. Line 18 and everywhere as applicable: Please, position ɛ as a superscript.
2. Line 36: plasma of patients, suffering from diseases. Diseases themselves do not have plasma
3. Line 62: provide please molar concentration for BSA in incubations
4. Line 68: “as shown in Table 1” should not be a separate sentence
5. Lines 86-87: somehow doubled information. You can put A,B and C already in the first sentence in the Figure legend
6. Line 228 and everywhere as applicable: D-glucose: D needs to be italic or small capital.
7. Line 269 and everywhere as applicable: please, control the presence of space between values and following directly after units.
8. Line 268: I did not find information about pre-activation of the cells. Is this procedure meant: “After several washes, the pre-treatment MNC were then acticated with anti-CD3+anti-CD28 antibodies in the presence of different concentrations of AGE for 24 h”? If yes, than it is too unprecise.
9. Line 275 and everywhere as applicable: replace everywhere µl with µL
10. Line 132: what do you mean as “concentrations of AGEs”?
Author Response
Answers to Reviewer [2]:
[Major remarks]:
Question (1): English needs to be improved!
Answer: We will ask MDPI publisher for English editing to improve the writing.
Question (2): The largest problem of this work: Why CML is taken for characterization of glycation levels? Why only CML? Why modifications at arginine residues are not considered? To some extent, proteomics studies could provide some information, but these data are not shown! This issue need to be comprehensively addressed and improved. The authors also define the sample in a strange way.
Answer: Thank to the Reviewer for raising these crucial questions to improve the quality of our manuscript. In general, AGEs include Ne-[carboxymethyl]lysine (CML), Ne-[carboxyethyl]lysine (CEL), imidazolone, methyl-glyoxal-lysine dimer (MOLD), glyoxal-lysine dimer (GOLD), pyrraline and pentosidine etc. In the present study, we only detected CML as surrogate of BSA-AGE detected by LC/MS proteomics analysis. Since quantitation of CML can be achieved by commercially available ELISA kit in addition to the detection of number and % of amino acid glycation by LC/MS as shown in Table 1. Un-expectedly, the glycation of arginine is modest and no glycation of cysteine was found. In addition, many authors have demonstrated that CML is an important component of BSA-AGE correlation with different diseases including cardiovascular complications (Atherosclerosis 221: 387-96, 2012), decreased insulin secretion from pancreatic b-cells (Am J Physiol Endocrinol Metab 309: E829-39, 2015), increased mortality in ICU (PLoS ONE 11: e0160893, 2016) and breast cancer (Histochem Cell Biol 147: 625-34, 2017). These data may indicate that CML measurement is quite useful in the clinical applications.
Question (3): Line 16 and everywhere as applicable: correct SI unit for concentration is mol/L, but not M. Please change everywhere appropriately. And think about spaces between values and units.
Answer: It is absolutely our carelessness. We have already corrected them everywhere in text.
Question (4): Line 25: What do you define as “AGE” here? AGE-modified protein? Needs to be precisely defined!
Answer: Thanks for this valuable comment. Here, we define AGE as glycation or glycoxidation of macromolecules including proteins, lipids, nucleic acids or glycoproteins. BSA-AGE is defined as AGE-modified protein used in the text. Accordingly, we have changed the title of the manuscript to “Advanced glycation-modified bovine serum albumin suppressed Th1/Th2 cytokine but….”. We will also carefully use the terminology in text.
Question (5): Introduction appears too short. It needs to be more detailed and to more comprehensively disclose interactions of AGEs and ageing.
Answer: Thanks for this valuable suggestion. We have already added two paragraphs related the AGEs and ageing in the “Introduction” section for more comprehension. These statements are shown below.
Paragraph (I): It is conceivable that macromolecular damage and biochemical changes may occur in physiological ageing and age-related disorders including diabetes, cataract, Alzheimar’s disease, amyloidosis, atherosclerosis and Parkinson’s disease [1]. Accordingly, ageing and age-related disorders can be attributed to the process of macromolecular glycation as a common event. Semba et al. [2] concluded that accumulation of AGE-modified macromolecules either ingested in foods or up-regulated by inflammation may accelerate the multisystem function decline in ageing and therefore contribute to the ageing phenotypes. Simm A [37] further demonstrated that glycation can modify the structure and function of intracellular and extracellular matrix expressions to cause tissue stiffness but long-lasting inflammatory response.
Paragraph (II): Several contributing factors such as changes in the innate and adaptive immune responses, chronic antigen stimulation, and the appearance of endogenous macromolecule and senescent cells can be found in blood as the potential contributory factors for inflamm-aging [40].
Question (6): I generally do not understand how ELISA was done here. Protocol and antibody description are necessary.
Answer: As pointed-out by the Reviewer, we measured the CML concentration in different incubation days of BSA-AGE by ELISA kit. The kit is obtained from CycLex CML/Ne-(carboxymethyl)lysine ELISA kit developed by CircuLex company for quantitative measurement of CML-adducts in mammalian serum, plasma, tissue extract and other biological media except rodent specimen. We have already added the methodology of competitive EIA in section of 4.2.3 (line 287):
“Briefly, biochemically modified CML-BSA was pre-coated in microwells. Standards or samples and anti-CML adduct monoclonal antibody MK-5A10 is added for competition. After incubation, the amount of CML-BSA was calculated by the standard curve”.
Question (7): Section 4.1: with the progression of Maillard reaction, original protein solution typically becomes a bit viscous. That is why an important question-did you shake incubation with HAS?
Answer: As Reviewer’s comment, we incubated BSA and glucose at 370C in 5% CO2-95% air after 45 days. We noted that fluid began to become a little bit viscous. We did not shake the mixture but only closely observe the color change. Increased molecular weight of this viscous BSA-AGE solution can be found in 10% SDS-PAGE as shown in Fig.1. We have added a phrase of “with a bit viscous” in line 278 in the new version.
Question (8): Section 4.2.4: the provided description of proteomics experiment is not sufficient. Please provide exact and detailed procedure for sample preparation, measurements and data interpretation.
Answer: We sent the 3 samples (Day-30, day-130 and day-180) to Mithra Biotechnology Inc. for GC/MC proteomic detection of BSA glycation. We have attached the complete report file from Mithra Biotechnology, Inc. as Supplement 2. The detailed sample preparation, measurements and data interpretation are all included in this “Glycation site identification report”.
Question (9): Section 4.2.5: the information, how lectin-binding test was done is missing.
Answer: It is really our carelessness for missing this important methodology information in lectin-binding assay in Section 4.2.5 (line 305). We have already added the statements of the information in this section for more clarity below:
“Briefly, various BSA-AGEs were loaded in 10% SDS-PAGE for electrophoretic detection of lectin-binding. After electrotransferred to nitrocellulose paper, the dispersed molecules were reacted with digoxigenin-labeled lectin and then alkaline phosphatase-conjugated anti-digoxigenin antibody. The complexes were finally color development by staining solution containing NBT/BCIP solution”.
Question (10): Section 4.5: too detailed with primers. Put primer sequences as a table in supplement.
Answer: We have already deleted the cytokine oligonucleotide pair primers from the old text. A suggested by the Reviewer, a new table showing the detailed oligonucleotide pair primers is attached as Supplement 3.
Question (11): Please make a list of abbreviation-it is really difficult without such a list in the case when results stand before methods.
Answer: It is really a valuable suggestion to make an abbreviation list after section of “Introduction”. We have already provided it in the revised version as shown below:
List of abbreviations:
AGE: advanced glycation end-products
BSA: bovine serum albumin
CML: Ne-(carboxymethy)-lysine
CEL: Ne-(caroxyethyl)-lysine
EC: endothelial cell
Gal: galactosamine
GlcNAc: N-acetyl-glucosamine
IL: interleukin
Inflamm-aging: inflammation in aging
IFN-g: interferon-g
MNC: mononuclear cell
Mf: monocyte/macrophage
PSS: progressive systemic sclerosis
RA: rheumatoid arthritis
RNS: reactive nitrogen species
ROS: reactive oxygen species
SLE: systemic lupus erythematosus
Th: helper T cell
TNF-a: tumor necrosis factor-a
Question (12): Line 11: what do you mean as “different AGEs”? You quantify only CML-I would not expolate it to all AGEs.
Answer: We totally agree with Reviewer’s viewpoint. As a matter of fact, only measuring BSA-CML is not enough since BSA-AGEs may include CML, CEL and the other glycation end-products was explained in the major remarks question (2). Among these AGEs, CML is crucial for the functional activities of BSA-AGE in this study. Therefore, we have changed different AGEs into “BSA-AGE” in text for more clarity.
[Miner remarks]:
Question (1): Line 18 and everywhere are applicable: please position e as a superscript.
Answer: Thanks for the suggestion. We have already corrected the subscript “Ne” to the superscript “Ne” everywhere in text.
Question (2): Line 36: plasma of patients suffering from diseases. Diseases them- selves do not have plasma:
Answer: Sorry for this grammatical error. We have already changed “plasma of aging…” to “ the plasma obtained from aged group…” for more clarity in line 38.
Question (3): Line 62: provide please molar concentration for BSA in incubation.
Answer: We have put molar concentration of BSA (0.0746 mol/L) in line 96 and everywhere in text.
Question (4): Line 68: “as shown in Table 1” should not be a separate sentence.
Answer: It is our typing error. We have corrected it to be “ As shown in Table 1, the amount of CML in BSA-AGE increased in parallel to…” in line 103 in the revised version.
Question (5): Lines 86-87: Somehow doubled information. You can put A, B and C already in the first sentence in the figure legend.
Answer: Thanks for informing us the duplicate descriptions in Figure 1 legend. We have put (A), (B), and (C) in the first sentence of the legend to prevent duplicate description in line 123.
Question (6): Line 228 (line 231?) and everywhere as applicable: D-glucose: D needs to be italic or small capital.
Answer: We obey your suggestion to change “D-glucose” into italic form “D-glucose” in line 272.
Question (7): Line 269 (line 271?) and everywhere as applicable: please control the presence of space between values and following directly after units.
Answer: Thanks for the suggestion. We have corrected them by a space between each value and unit everywhere in text.
Question (8): Line 268 (line 270?): I did not find information about pre-activation of the cells. Is this procedure means: After several washes, the pre-treated MNC were then activated with anti-CD3+ anti-CD28 antibodies, in the presence of different concentrations of AGE for 24 h? If yes, than it is too un-precision.
Answer: As Reviewer points out the statements in old line 270 is vague and not precision. In the revised version, we changed the statements as “Anti-CD3+anti-CD28-preactivated human MNC (2x106/mL) at 370C were incubated with 1 mg/mL… for 2 hours. The total RNAs were then extracted for detection of Th1 (denoted by IL-2 and IFN-g) and Th2 (denoted by IL-10) cytokine mRNA expression. In addition, the pre-activated cells were incubated with different amounts of BSA-AGE or 24 h. The culture supernatants were collected for IL-6 protein determination by ELISA” in Section 4.4. line 322.
Question (9): Line 275 and everywhere as applicable. Replace everywhere ml with mL.
Answer: Thanks for the suggestion. We have revised them in the new version.
Question (10): Line 132: What do you mean as “concentrations of AGEs”?
Answer: For more precision, we have changed the phrase of “different concentrations of AGEs” to “different amounts of BSA-AGE” in line 166.

Round 2
Reviewer 1 Report
The authors revised the manuscript appropriately according to the comments, and therefore the revised manuscript could be acceptable for the publication in its present form.
Author Response
Answers to Reviewer [1]:
Question:
The authors revised the manuscript appropriately according to the comments, and therefore the revised manuscript could be acceptable for the publication in its present form.
Answer:
We are sincerely grateful for reviewer’s kind decision and inspiring advise. We have learned a lot from reviewer’s suggestion and would keep working on the field of carbohydrate chemistry.

Reviewer 2 Report
Dear Sir or Madam,
the authors have done a good job in correcting their manuscript. However, something is still to do:
1. The title would be better to leave as: “Advanced glycation end products of bovine serum albumin….”. Аnd I don’t really understand „but“ in the title – I think, still English problems
2. Line 36: end products – two words
3. Line 67: what is “inflamm-ageing”?
4. Line 74: amino acid
5. Line 84: not LC/MS, but LC-MS.
6. Line 118: “Glycation” is singular – you probably mean “sites of glycation”?
7. Line 161: you mean alterations in lectin binding?
8. Somewhere in text needs to be explained , why the glycation mixtures were not shaken – it might basically result in loss of reproducibility. Why shaking was omitted?
9. The most problems: description of proteomics methods is missing – it needs to be provided, wherever did it. The description in lines 270-276 is insufficient.
Author Response
Answers to Reviewer [2]:
Question (1) The title would be better to leave as: “Advanced glycation end products of bovine serum albumin….”. Аnd I don’t really understand „but“ in the title – I think, still English problems
Answer: Thank to reviewer’s suggestion. We will adjust our title as the reviewer’s suggestion.
Question (2) Line 36: end products – two words
Answer: It is absolutely our carelessness. We have already corrected and all other mentioned in our article.
Question (3) Line 67: what is “inflamm-ageing”?
Answer: Thank to the Reviewer for raising these crucial questions to improve the quality of our manuscript. We have borrowed the concept of “inflamm-ageing” from endocrinologist and inflamm-ageing means “chronic, sterile, low grade inflammation developed during aging”. This concept is completely discussed in the article investigated by Franceschi, C., et al., publishing in Nat Rev Endocrinol, 2018. 14(10): p. 576-590., entitled as Inflammaging: a new immune-metabolic viewpoint for age-related diseases. We will also add this article as our reference
Question (4) Line 74: amino acid
Answer:Thanks for the suggestion. We have already obeyed as adjusting the term “albumin” to “amino acid”
Question (5) Line 84: not LC-MS, but LC-MS.
Answer: Thank you for your kindly suggestion. We have already corrected everywhere in the text.
Question (6) Line 118: “Glycation” is singular – you probably mean “sites of glycation”?
Answer: Thank reviewer for the suggestion, We have already replace “glycation” to “sites of glycation” in line 118.
Question (7) Line 161: you mean alterations in lectin binding?
Answer: Thank to reviewer for point out this statement not precise. Yes, we mean alterations in lectin binding.
Question (8) Somewhere in text needs to be explained , why the glycation mixtures were not shaken – it might basically result in loss of reproducibility. Why shaking was omitted?
Answer: Thank reviewer to point out this very important point. We will be very willing to produce AGE with gently shaken during incubation in the following study. Before initiation this current study, we have searched literature about formation method of AGE-modified BSA. All publications mentioned concentration of albumin, glucose (or type of glucose), incubation period and incubation environment. However, the shaking method was not mentioned in most articles. Therefore, we considered shaking during AGE incubation is not routinely done.
Question (9) The most problems: description of proteomics methods is missing- it
needs to be provided, wherever did it. The description in lines 270-276 is
insufficient.
Answer: Thank to the reviewer point out this part. We have replace the LC-MS method the detail of LC-
MS experiment details in 4.2.4 of the text.
“We have Ammonium bicarbonate, DTT, IAM and FA purchased from Sigma-Aldrich(St. Louis, MO, USA). ACN was purchased from Spectrum(New Jersey, USA). Trypsin and Chymotrypsin were purchased from Promega(Madison, WI. USA). RapiGest SF was purchased from Waters. TFA was purchased from Alfa Aesar(Ward Hill, MA, USA).
Advanced glycation end products of bovine serum albumin solution was buffer exchanged with 50 mM ammonium bicarbonate using a 10 kDa molecular weight cutoff filter. The three samples wereadded with 1% RapiGestSF Surfactant to a final concentration of 0.1%, reduced with 5 mM DTT at 60 °Cfor 30 min, and then alkylated with 15 mM IAM in dark at room temperature for 30 min. The resulting protein was digested withtwo enzymes with the following digestion conditions: (1) Trypsin digestion at 37 °Covernight (protein: enzyme = 50:1), (2) Chymotrypsin digestion at room temperature overnight (protein:enzyme =50:1). After digestion, each sample was acidified with 0.5% (v/v) TFA as the final concentration, which was performed at 37 °C for30 min and then centrifuged at 14000 rpm for 30 min to precipitate the hydrolytic RapiGest SF by-product.
The digested samples were analyzed with Q-Exactive mass spectrometer (Thermo Scientific, Waltham, MA, USA) coupled with Ultimate 3000 RSLC system (Dionex, Sunnyvale, CA, USA). The LC separation was performed using a C18 column (Acclaim PepMap RSLC, 75 μm x 150 mm, 2 μm, 100 Å). We use Mobile phase A: 0.1% FA in H2O and Mobile phase B: 0.1% FA in 95% ACN. The flow rate is 0.25 μL/min. The gradient setting is 0 min-1%, 5.5 min-1%, 45 min-25%, 48 min 60%, 50 min 80%, 65 min-1%, 70 min-1%.
Full MS scan was performed with the range of m/z 150-2000, and the ten most intense ions from MS scan were subjected to fragmentation for MS/MS spectra. Raw data were processed into peak lists by Proteome Discoverer 1.4 for Mascot database search.
Database search was performed with Mascot 2.4.1. with the instument type as ESI-TRAP. The parameters used were as follows:
We use BSA Database: BSA. TheFixed modifications is Carbamidomethyl (C) and the variable modifications as Glycation (K), Glycation (A) and Glycation (C). The peptide mass tolerance is 10 ppm and the fragment mass tolerance is 0.05 Da. The ion cut-off score is 13.”
